# HDAC8-Selective Inhibition by PCI-34051 Enhances the Anticancer Effects of ACY-241 in Ovarian Cancer Cells

**DOI:** 10.3390/ijms23158645

**Published:** 2022-08-03

**Authors:** Ji Yoon Kim, Seung Yoon Han, Jung Yoo, Go Woon Kim, Yu Hyun Jeon, Sang Wu Lee, Jongsun Park, So Hee Kwon

**Affiliations:** 1College of Pharmacy, Yonsei Institute of Pharmaceutical Sciences, Yonsei University, Incheon 21983, Korea; jiyoon1323@yonsei.ac.kr (J.Y.K.); battalion1@yonsei.ac.kr (S.Y.H.); jungy619@yonsei.ac.kr (J.Y.); gowoon@yonsei.ac.kr (G.W.K.); uhyun953@yonsei.ac.kr (Y.H.J.); tkddn407@yonsei.ac.kr (S.W.L.); 2Department of Pharmacology, College of Medicine, Chungnam National University, Daejeon 35015, Korea; insulin@cnu.ac.kr; 3Metabolic Syndrome and Cell Signaling Laboratory, Department of Medical Science, Institute for Cancer Research, College of Medicine, Chungnam National University, Daejeon 35015, Korea

**Keywords:** epigenetics, HDAC6, HDAC8, ACY-241, PCI-34051, ovarian cancer

## Abstract

HDAC6 is overexpressed in ovarian cancer and is known to be correlated with tumorigenesis. Accordingly, ACY-241, a selective HDAC6 inhibitor, is currently under clinical trial and has been tested in combination with various drugs. HDAC8, another member of the HDAC family, has recently gained attention as a novel target for cancer therapy. Here, we evaluated the synergistic anticancer effects of PCI-34051 and ACY-241 in ovarian cancer. Among various ovarian cancer cells, PCI-34051 effectively suppresses cell proliferation in wild-type p53 ovarian cancer cells compared with mutant p53 ovarian cancer cells. In ovarian cancer cells harboring wild-type p53, PCI-34051 in combination with ACY-241 synergistically represses cell proliferation, enhances apoptosis, and suppresses cell migration. The expression of pro-apoptotic proteins is synergistically upregulated, whereas the expressions of anti-apoptotic proteins and metastasis-associated proteins are significantly downregulated in combination treatment. Furthermore, the level of acetyl-p53 at K381 is synergistically upregulated upon combination treatment. Overall, co-inhibition of HDAC6 and HDAC8 through selective inhibitors synergistically suppresses cancer cell proliferation and metastasis in p53 wild-type ovarian cancer cells. These results suggest a novel approach to treating ovarian cancer patients and the therapeutic potential in developing HDAC6/8 dual inhibitors.

## 1. Introduction

Epithelial ovarian cancer (EOC) is the seventh most common type of cancer worldwide in women and is the most fatal gynecologic cancer, with a 5-year survival rate of roughly 50%. Early detection of ovarian cancer is difficult owing to the vagueness of its symptoms. In fact, more than 70% of women are diagnosed with Stage II or IV [1]. While early-stage presentation has a 5-year survival rate of 92%, the survival rate drops to 29% in late-stage presentation [2]. Thus, effective diagnostic methods and treatment plans are crucial in overcoming the disease. Standard treatment strategies involve aggressive cytoreductive surgery along with platinum/taxane chemotherapy. However, more than 70% of patients with advanced disease develop recurrence within the first 5 years [3]. Accumulated toxicities from chemotherapy present further complications for such treatment in overcoming ovarian cancer.

Aberrant epigenetic alteration is prevalent in tumorigenesis and various epigenetic regulators are currently being evaluated as potential targets for cancer treatment. Histone deacetylases (HDACs) represent a set of eighteen enzymes that catalyze the removal of acetyl groups from ε-amino lysine of histones and other non-histone substrates. HDACs are well-studied cancer targets that are overexpressed in various types of cancer and are associated with advanced disease and poor outcomes in patients [4,5,6,7]. Among the four subclasses that comprise the HDAC enzyme family, HDAC6 is a member of the Class IIb HDAC family and mediates the deacetylation of not only histone, but also non-histone proteins such as α-tubulin, cortactin, p53, and heat shock protein 90 (HSP90) [8]. Abnormal expression of HDAC6 is correlated with neurodegenerative diseases, cardiovascular diseases, and cancer [9,10,11,12]. Specifically, HDAC6 participates in cell proliferation, metastasis, invasion, and mitosis in tumors, and is overexpressed in ovarian cancer, bladder cancer, lung cancer, and colon cancer [12,13,14,15]. Overexpression of HDAC6 has also recently been presented as a favorable prognostic marker of ovarian cancer [16]. Therefore, many efforts have been focused on developing inhibitors that target HDAC6. Among numerous HDAC inhibitors (HDACis) currently under clinical trials, five have been approved by the Food and Drug Administration (FDA) thus far [17]. However, because of undesirable off-target toxicities of pan-HDACi, HDAC6-selective inhibitor (HDAC6i) is gaining more attention. Citarinostat (ACY-241) is a second-generation HDAC6 selective inhibitor that shows high selectivity and improved solubility over structurally related ricolinostat (ACY-1215). An ongoing investigation is being conducted to evaluate the antitumor effect of HDAC6i alone or in combination with other drugs [18,19,20,21,22,23]. 

HDAC8 is a member of the Class I HDAC family. The deacetylating activity of HDAC8 on histones remains yet to be uncovered. However, a large number of non-histone substrates such as structural maintenance of chromosomes 3 (SMC3), estrogen-related receptor alpha (ERRα), and p53 have been recognized as HDAC8 target proteins. HDAC8 expression is involved in various diseases including Cornelia de Lange syndrome (CdLS), viral infections, schistosomiasis, and cancer [24]. HDAC8 is upregulated in breast cancer, urothelial cancer, hepatocarcinoma, and acute lymphatic leukemia, and is associated with poor prognosis and low overall survival rates in neuroblastoma [25,26,27,28,29]. As the most recently identified class I HDAC, however, a comprehensive understanding of HDAC8 in cancer remains unclear. Nevertheless, HDAC8 is an attractive therapeutic target and several HDAC8 selective inhibitors (HDAC8is) have been developed [30]. PCI-34051 is a selective HDAC8 inhibitor with 200-fold selectivity compared with other HDACs and has been found to be effective in T-cell lymphoma and leukemia [31]. 

Here, we evaluated the anti-cancer effect of PCI-34051 and ACY-241 in ovarian cancer. We found that inhibition of HDAC8 had a greater anti-proliferative effect in ovarian cancer cells harboring wild-type p53 rather than mutant p53. Our work also identified enhanced p53 acetylation levels and synergistic anti-cancer effects when HDAC8i and HDAC6i were treated in combination. These results suggest a potential combination therapy in treating ovarian cancer.

## 2. Results

### 2.1. HDAC8 Inhibitor, PCI-34051, Suppresses Cell Growth and Reduces Cell Viability in p53 Wild-Type Ovarian Cancer Cells

Various studies have analyzed the anti-cancer effect of the HDAC6 inhibitor, ACY-241, and its combination effect with standard platinum/taxane chemotherapy in different types of cancer, including ovarian cancer [18,20,32,33]. However, no study has been conducted on the HDAC8 inhibitor, PCI-34051, in ovarian cancer so far. To examine the anti-cancer effect of PCI-34051, we performed a CCK-8 assay to analyze cell growth and viability in various ovarian cancer cell lines. To our surprise, PCI-34051 showed a greater anti-proliferative effect in ovarian cell lines with wild-type p53 (Figure 1A–D) compared with those with mutant p53 (Figure 1E–H). While the half-maximal inhibitory concentration (IC_50_) values for wild-type p53 ovarian cancer cell lines, TOV-21G and A2780, were 9.73 μM and 28.31 μM, respectively, these values were greatly elevated in mutant p53 ovarian cancer cell lines, COV318 and COV362, with IC_50_ values of 127.6 μM and 120.4 μM, respectively (Table 1). Similar to IC_50_ values, mutant p53 ovarian cancer cell lines showed greater half-maximal growth inhibition concentration (GI_50_) values than wild-type p53 cell lines. Thus, we concentrated our investigation on wild-type p53 ovarian cancer. 

### 2.2. ACY-241 and PCI-34051 Treatment Synergistically Suppresses Cell Growth and Reduces Cell Viability in p53 Wild-Type Ovarian Cancer Cells

In order to verify the synergistic anti-cancer effect of ACY-241 and PCI-34051 in wild-type p53 ovarian cancer, we used the CCK-8 assay and examined the combination index (CI) calculated based on the Chou and Talalay method (Figure 2A,B) [34]. The CI value less than, equal to, or more than 1 indicates synergism, additive effect, or antagonism, respectively. Cell viability was significantly decreased when ACY-241 and PCI-34051 were treated in combination at a dose ratio of 1:5 compared with when they were treated separately. CI values ranged from 0.57 to 1.17 in TOV-21G and from 0.65 to 1.29 in A2780, suggesting the synergistic antiproliferative effect of ACY-241 and PCI-34051 in wild-type p53 ovarian cancer cells. Next, we performed a colony formation assay to study the long-term effect of ACY-241 and PCI-34051 (Figure 2C,D) in ovarian tumorigenesis. Cells were incubated with ACY-241 and PCI-34051 alone or in combination. Colony formation was significantly reduced when the inhibitors were treated in combination rather than when they were treated alone. These findings indicate that inhibition of HDAC6 and HDAC8 synergistically suppresses both short- and long-term ovarian cancer cell proliferation.

### 2.3. ACY-241 and PCI-34051 Treatment Synergistically Induces Apoptosis in p53 Wild-Type Ovarian Cancer Cells

HDACi, like most anticancer drugs, is known to induce tumor cell death by apoptosis [35]. Here, we performed immunoblotting to examine whether ACY-241 and PCI-34051 would synergistically induce apoptosis (Figure 3A,B). Combination treatment of ACY-241 and PCI-34051 increased cleaved poly(ADP-ribose) polymerase (PARP) and cleaved caspase-3 in TOV-21G cells. Additionally, when ACY-241 and PCI-34051 were treated together, pro-apoptotic protein, Bak, was synergistically increased in TOV-21G and A2780 cells, while anti-apoptotic proteins, Bcl-XL and XIAP, were significantly reduced in both cell lines. To further confirm the apoptotic effect of ACY-241 and PCI-34051, we performed an apoptosis assay using Annexin V/PI double staining (Figure 3C,D). The percentage of apoptotic cells was synergistically increased in both A2780 and TOV-21G cells when the inhibitors were treated in combination compared with when they were treated as single agents. Together, these results show that treatment of ACY-241 and PCI-34051 has a greater apoptotic effect in combination, contributing to the synergistic anti-proliferative effect confirmed in the cell viability assays. 

### 2.4. ACY-241 and PCI-34051 Treatment Synergistically Inhibits Cell Migration in p53 Wild-Type Ovarian Cancer Cells

Several studies have revealed that HDAC6 and HDAC8 both play a crucial role in cell migration and invasion [36,37]. Thus, we performed immunoblotting and analyzed the expression levels of several metastatic markers after treating the cells with 3 µM ACY-241 and 20 µM PCI-34051 (Figure 4A,B). Interestingly, EMT-related proteins, TWIST1, MMP-9, and ZEB1, were significantly downregulated when the two inhibitors were treated in combination. We further demonstrated the synergistic effect of ACY-241 and PCI-34051 in inhibiting metastasis by performing the wound-healing assay and transwell migration assay in TOV-21G and A2780 cells (Figure 4C–F). Wound-healing assay results revealed that the migration ability was greatly diminished when cells were treated with ACY-241 and PCI-34051 compared with when they were treated individually. Similarly, transwell analysis showed that the number of migrated cells was greatly reduced in combination treatment compared to single treatment. Together, these results suggest that ACY-241 and PCI-34051 synergistically weaken the migration capacity of p53 wild-type ovarian cancer cells.

### 2.5. ACY-241 and PCI-34051 Treatment Synergistically Enhances p53 Stability in p53 Wild-Type Ovarian Cancer Cells 

p53 acetylation is known to be strongly correlated with protein stabilization and activation of p53 [38]. Furthermore, various studies have demonstrated that HDAC inhibition can increase p53 lysine acetylation and evoke its tumor-suppressive functions [39]. Nevertheless, the molecular response following HDACi treatment may vary depending on the class of HDACi and the tumor type. In previous research, HDAC6 inhibition was shown to increase acetyl-p53 at lysine 381 (K381) [40]. Here, we analyzed the expression pattern of acetyl-p53 (K381) upon HDAC8 inhibition (Figure 5A). PCI-34051 increased acetyl-p53 (K381) in a dose-dependent manner in p53 wild-type ovarian cancer cells. 

Next, we evaluated whether HDAC6 and HDAC8 inhibitors could synergistically induce p53 acetylation (Figure 5B,C). Acetyl-α-tubulin and acetyl SMC3 were used as positive controls to confirm the enzymatic inhibitory activities of ACY-241 and PCI-34051, respectively. ACY-241 increased the acetylation of α-tubulin and PCI-34051 increased the acetylation of SMC3. PCI-34051 did not cause detectable α-tubulin acetylation when treated alone, as demonstrated in various studies [31,41]. However, combination treatment of ACY-241 and PCI-34051 significantly increased α-tubulin acetylation, suggesting a cooperative role of PCI-34051 in acetylating α-tubulin when treated along with ACY-241. Co-treatment of ACY-241 and PCI-34051 did not show a dramatic increase in p53 levels compared to single treatment. However, the levels of acetyl-p53 (K381) were greatly elevated upon combination treatment, illustrating enhanced stabilization and activation of p53. Protein expression levels of p21, a cyclin-dependent kinase inhibitor (CKI) and a well-known p53 target gene, correlated with acetyl-p53 levels and showed a significant increase when the agents were treated together. Collectively, our studies demonstrated that inhibition of HDAC6 and HDAC8 through selective inhibitors ACY-241 and PCI-34051 synergistically increases the stability of p53, which may contribute to the enhancement in apoptosis and inhibition of metastasis in wild-type p53 ovarian cancer cells.

## 3. Discussion

In this study, we demonstrate, for the first time, the anti-cancer effect of HDAC8 selective inhibitor, PCI-34051, in ovarian cancer. While ovarian cancer cells with wild-type p53 have a significant anti-proliferative effect upon PCI-34051 treatment, ovarian cancer cells harboring mutant p53 show minimal anti-proliferative effects. Furthermore, we identified that HDAC8 inhibition through PCI-34051 increases acetyl-p53 at K381 in wild-type p53 ovarian cancer cells. We further analyzed the synergistic effects of PCI-34051 with HDAC6 selective inhibitor, ACY-241. Combination treatment of ACY-241 and PCI-34051 synergistically inhibits cell growth, induces apoptosis, and suppresses metastasis in wild-type p53 ovarian cancer, A2780 and TOV-21G. 

Among various mutations present in ovarian carcinogenesis, mutations in *TP53* that encode tumor suppressor p53 protein have been identified to be one of the most predominant genetic alterations in EOC [42]. In its wild-type form, p53 is a master regulator of diverse cellular processes such as cell apoptosis, cycle arrest, metabolism, and metastasis [43]. However, oncomorphic mutations give rise to mutant p53, which undermines tumor-suppressive functions of p53 and promotes cancer cell survival and tumor progression [44]. Thus, while enhancing the protein level of wild-type p53 would provide anti-tumor effects in wild-type p53 ovarian cancer, suppressing the expression of mutant p53 would have therapeutic benefits in mutant p53 ovarian cancer. When mutant p53 harboring cells, COV318 and COV362, were treated with PCI-34051, this had minimal effects on p53 expression and its acetylation level (Appendix A). This is in correlation with the cell growth and viability assay, which demonstrated that PCI-34051 had limited effects on cell proliferation of ovarian cancer cells with mutant p53. Furthermore, when ACY-241 and PCI-34051 were treated in the mutant p53 ovarian cancer cell, COV362, neither acetyl-p53 (K381) nor acetyl-α-tubulin were synergistically upregulated (Appendix A). Moreover, pro-apoptotic markers and anti-apoptotic makers did not show significant changes in their expression pattern (Appendix A). Further investigation is required to confirm whether the lack of a synergistic anti-cancer effect was indeed due to the p53 mutation status of the ovarian cancer cells. Nonetheless, the mutation status of various proteins, especially p53, should be thoroughly evaluated when designing therapeutic interventions to enhance the efficacy of molecularly targeted treatments. 

Previous studies have elucidated the individual effects of HDAC6i and HDAC8i in promoting p53 acetylation and its transcriptional activation [40,45]. Specifically, HDAC6 inhibition promotes p53 acetylation at K381 and K382, thereby inducing p53-mediated apoptosis in colorectal cancer (CRC) cells [40]. HDAC8 inhibition, through the selective inhibitor, 22d, enhances p53 activation by enhancing p53 at K382 in leukemia stem cells (LSCs) [46]. Thus, we analyzed the acetylation pattern of p53 at K381 and K382. However, acetyl-p53 at K382 was not detectable in ovarian cancer cells, A2780 and TOV21G (data not shown). Nonetheless, here, we demonstrated that PCI-34051 increases acetyl-p53 at K381 in a dose-dependent manner. Furthermore, in combination with ACY-241, PCI-34051 treatment greatly enhanced acetyl-p53 (K381) levels. Together, these results suggest that inhibiting both HDAC6 and HDAC8 will synergistically increase acetylation of p53 at K381, thereby promoting p53 stability and activating tumor-suppressive functions of wild-type p53.

Microtubules are cytoskeletal polymers that perform important cellular functions such as protein trafficking, cell cycle, and cell migration [47]. However, the role of tubulin acetylation in cancer remains controversial. While α-tubulin acetylation suppresses cell migration and invasion in lung cancer, it shows the opposite effect in breast cancer [48,49]. The different cellular functions of acetyl-α-tubulin in cancer cells may vary depending on the cellular environment and innate properties of distinct cell types [50]. Various reports have verified HDAC6 as a key α-tubulin deacetylase, while HDAC8 has recently been recognized as a deacetylase of α-tubulin [51]. Here, we demonstrated that the combination treatment of ACY-241 and PCI-34051 synergistically increases acetyl-α-tubulin, and thus may take part in the suppression of metastasis observed in our results. 

Ongoing studies have been conducted to uncover aberrant epigenetic modifications in cancer development and various efforts have been devoted to finding epigenetic inhibitors that can reverse these abnormal changes. However, because of the complexity of epigenetic regulations, recent investigations have concentrated on finding combination strategies involving two or more drugs [52]. Combination therapy is an attractive therapeutic strategy and has shown anti-cancer benefits in reducing tumor growth and metastasis, arresting the cell cycle, and inducing apoptosis [53]. Various ongoing clinical trials are evaluating the effect of combination therapy targeting epigenetic events [54]. ACY-241, a second-generation HDAC6 selective inhibitor, has been studied in combination with various agents in multiple cancers such as paclitaxel in solid tumor models; anti-PD-L1, pomalidomide, and daratumumab in multiple myeloma (MM); JQ1 in head and neck cancer; nivolumab in advanced non-small cell lung cancer; and erlotinib in pancreatic cancer [18,19,20,21,22,23,55]. On the other hand, the HDAC8 selective inhibitor, PCI-34051, has been shown to be effective when treated with KPT-2974 or cytarabine in acute myeloid leukemia and crizotinib or retinoic acid in neuroblastoma [56,57,58,59]. Although various studies have revealed synergistic anti-cancer effects when using HDACis with chemotherapeutic agents, therapeutic strategies targeting two epigenetic modulators have not been thoroughly investigated. In this study, we aimed to find an epigenetic target that would work synergistically with HDAC6i, which has been proven to be effective in ovarian cancer. HDAC8, which shows functional redundancy with HDAC6 and has recently emerged as an attractive anticancer target, was selected and tested in combination with ACY-241 for synergistic effects. 

Along with the efforts to find effective epigenetic drug combinations, several attempts have been made at developing dual inhibitors that can simultaneously inhibit two epigenetic targets such as UNC1999, which inhibits both EZH2 and EZH1; CM S-272, which targets G9a and DNMT1; and Corin, an LSD1/HDAC inhibitor [60,61,62]. Furthermore, TH34, an HDAC6/8/10 inhibitor, and droxinostat, an HDAC3/6/8 inhibitor, have demonstrated anti-cancer effects in several cancer types [63,64]. As our results suggest, dual inhibitors targeting both HDAC6 and HDAC8 may prove to be an effective anti-cancer agent as they would decrease the undesirable cytotoxic side effects from having to use multiple inhibitors while still providing the anti-cancer benefits of inhibiting HDAC6 and HDAC8. 

Further investigation is required to determine whether inhibition of HDAC6 and HDAC8 has synergistic anti-cancer effects in solid tumors beyond ovarian cancer. Our research was limited to ovarian cancer cells with wild-type p53. However, as p53 mutation is prevalent in EOC, future research should be conducted to elucidate alternative strategies that are effective against cancer cells with mutant p53. Overall, our results suggest that EOC patients with wild-type p53 may benefit from treatments combining HDAC6 inhibitor with HDAC8 inhibitor, and propose the potential benefit in developing an HDAC6/8 dual inhibitor.

## 4. Materials and Methods

### 4.1. Reagents

PCI-34051 and ACY-241 (Citarinostat) were obtained from CSNpharm (Arlington Heights, IL, USA). All agents were dissolved in DMSO (Sigma Chemical, St. Louis, MO, USA). The following antibodies were used: GAPDH (AP0066, Bioworld Technology, Bloomington, MN, USA), tubulin (sc-32293, Santa Cruz Biotechnology, Santa Cruz, CA, USA), Bak (sc-832, Santa Cruz Biotechnology), SMC3 (sc-376352, Santa Cruz Biotechnology), p53 (sc-126, Santa Cruz Biotechnology), p21 (sc-53870, Santa Cruz Biotechnology), PARP (551024, BD Biosciences, San Jose, CA, USA), XIAP (610716, BD Biosciences), MMP-9 (A0289, ABclonal Technology, Woburn, MA, USA), caspase-3 (#9662, Cell Signaling Technology, Danvers, MA, USA), Bcl-xL (#2762, Cell Signaling Technology), TWIST1 (#46702, Cell Signaling Technology), acetyl α-tubulin (T6793, Sigma-Aldrich, St. Louis, MO, USA), ZEB1 (HPA027524, Sigma-Aldrich), HDAC6 (A301-342A-M, Bethyl Laboratories, Montgomery, TX, USA), HDAC8 (ab187139, Abcam, Cambridge, UK), and ac-SMC3 (MABE1073, Millipore, Burlington, MA, USA).

### 4.2. Ovarian Cancer Cell Lines and CELL Culture

The human ovarian cancer cell line, TOV-21G (p53-WT), was purchased from American Type Culture Collection (ATCC). The human ovarian cancer cell lines A2780 (p53-WT), COV318 (p53-I195F), and COV362 (p53-Y220C) were purchased from the European Collection of Authenticated Cell Cultures (ECACC). TOV-21G and A2780 were cultured in Roswell Park Memorial Institute medium (RPMI) (Welgene, Daegu, Korea). COV318 and COV361 were cultured in Dulbecco’s modified Eagle’s medium (Sigma-Aldrich; St. Louis, MO, USA). Both media contained 10% fetal bovine serum (HyClone; GE Healthcare), 100 U/mL penicillin, and 100 µg/mL streptomycin (Gibco; Thermo Fisher Scientific, Inc., Waltham, MA, USA). Cell lines were maintained in 5% CO_2_ and 37 °C humidified incubators. Cells were subcultured every 3–4 days.

### 4.3. Cell Growth and Viability Assay

Cell growth and viability were determined by CCK-8 assay (WST-8; CCK-8 kit, CK04; Dojindo Molecular Technologies, Inc., Kumamoto, Japan). Cells were seeded in 96-well plates containing 130 μL medium (3 × 10^3^ cells per well). Following overnight incubation, PCI-34051 and ACY-241 were treated alone or in combination for 24 h, 48 h, and 72 h. Then, 13 μL of CCK-8 reagent was applied to each well for 3 h and a multimode microplate reader (Tecan Group, Ltd., Mannedorf, Switzerland) was used to measure the absorbance at 450 nm. The results were quantified relative to the control wells and presented as percentages. GraphPad Prism ver. 7.0 software (Graphpad Software, San Diego, CA, USA) was used to calculate the half-maximal inhibitory concentration (IC_50_) and half-maximal growth inhibition concentration (GI_50_).

### 4.4. Drug Combination Analysis

Cells were seeded in 96-well plates (3 × 10^3^ cells per well). Following overnight incubation, PCI-34051 and ACY-241 were treated at a constant ratio of 1:5, and then incubated for another 48 h. Cell viability was determined using the CCK-8 assay. Chou and Talay’s method was used to evaluate synergism between PCI-34051 and ACY-241 [34]. Calcusyn (Biofosft) was used to draw the fraction-affected (Fa) versus combination index (CI) plot. CI less than, equal to, and more than 1 implies synergy, additivity, and antagonism, respectively.

### 4.5. Apoptosis Assay

Cells were seeded in a 100 mm cell culture dish (1 × 10^6^ cells per plate). Following overnight incubation, cells were treated with ACY-241 3 μM and PCI-34051 20 μM alone or in combination for 48 h. Then, 1 × phosphate-buffered saline (PBS) was used to wash the cells, and trypsin and EDTA were used to detach the cells. Cells were pelleted, then resuspended using 1× binding buffer. Cells were stained for 15 min with 0.7 μL of Annexin V-fluorescein isothiocyanate (FITC) and 5 μL propidium iodide (PI) in the dark. Stained cells were diluted with 400 μL of 1× binding buffer (Annexin V-FITC Apoptosis Detection Kit; BD556547, BD Pharmingen San Diego, CA, USA). The stained cells were analyzed using a flow cytometer and BD FACSDiva software version 7 (both from BD Biosciences, San Jose, CA, USA).

### 4.6. Colony Formation Assay

Cells were seeded in a six-well plate (1 × 10^3^ cells per well). Following overnight incubation, PCI-34051 and ACY-241 were treated alone or in combination and incubated for another 7 days for A2780 and 14 days for TOV-21G cells. Cell colonies were fixed and stained using 1 mL of 0.05% crystal violet solution (Sigma-Aldrich; St. Louis, MO, USA) for 10 min. The number of colonies was counted and digital images of the colonies were taken with a camera.

### 4.7. Wound Healing Assay

Cells were seeded in a six-well plate (1 × 10^6^ cells per well). Following overnight incubation, a linear wound midline was created using a 200 μL pipette tip. The scratched cells were rinsed with serum-free medium to remove the cell debris and ACY-241 and PCI-34051 were treated alone or in combination and incubated for 24 h. The scratch closure was analyzed using an inverted microscope (Carl Zeiss AG, Feldbach, Switzerland) and imaged and analyzed with ImageJ software (National Institutes of Health, Bethesda, MD, USA).

### 4.8. Transwell Migration Assay

Cells were seeded in the upper chamber of the transwell insert plate (PET membrane, 8 μM pore, SPL 36224, Korea) in a 400 μL serum-free medium (1 × 10^5^ cells per well) and 500 μL of medium containing 10% FBS was placed in the lower chamber. Following overnight incubation, 3 μM ACY-241 and 20 μM PCI-34051 were added alone or in combination to the lower well. After 48 h incubation, cotton swabs were used to remove the non-migrated cells, while migrated cells were fixed and stained with 0.1% crystal violet solution containing 20% methanol. Light microscope with iSolution Lite (IMT i-Solution Inc., Vancouver, BC, Canada) was used to visualize the stained cells. Then, 10% acetic acid was used to solubilize the stained cells. A multimode microplate reader was used to measure the absorbance at 560 nm.

### 4.9. Western Blot

TOV-21G and A2780 cells were seeded in a six-well plate (5 × 10^5^ cells per well). Following overnight incubation, ACY-241 and PCI-34051 were treated alone or in combination for 24 h. Cells were washed with ice-cold 1× PBS and lysed with 100 µL lysis buffer (0.5% NP-40, 50 mM Tris-HCl (pH 7.4), 1 mM EDTA, 5 mM EGTA, 120 mM NaCl, 25 mM NaF, 25 mM glycerol phosphate, 1 mM PMSF, and 1 mM bezamidine). Bradford protein assay was used to measure the protein concentrations of cell lysates. Samples were prepared with a 5× sample buffer and subjected to SDS-PAGE on a polyacrylamide gel. Proteins were transferred onto a nitrocellulose membrane and blocked with 5% skim milk at room temperature for 1 h. The membranes were incubated with primary antibodies at 4 °C overnight. Membranes were washed three times with 0.1% Tween-20/PBS and incubated with anti-mouse or anti-rabbit secondary antibody coupled to HRP for 3 h at room temperature. Protein bands were visualized using the ECL Western blotting analysis system (Thermo Scientific Pierce, Waltham, MA, USA).

### 4.10. Statistical Analysis

GraphPad Prism software 7.0 (Graphpad Software, San Diego, CA, USA) was used to analyze the statistical significance of the experiments. All data presented are shown as means ± standard deviation (SD) of more than three independent experiments. Statistical significance was determined by unpaired two-tailed Student’s *t*-test and two-way analysis of variance (ANOVA) with post-hoc analysis using Turkey’s multiple comparison test. *p*-values < 0.05 were considered to indicate a statistically significant difference.

## Figures and Tables

**Figure 1 ijms-23-08645-f001:**
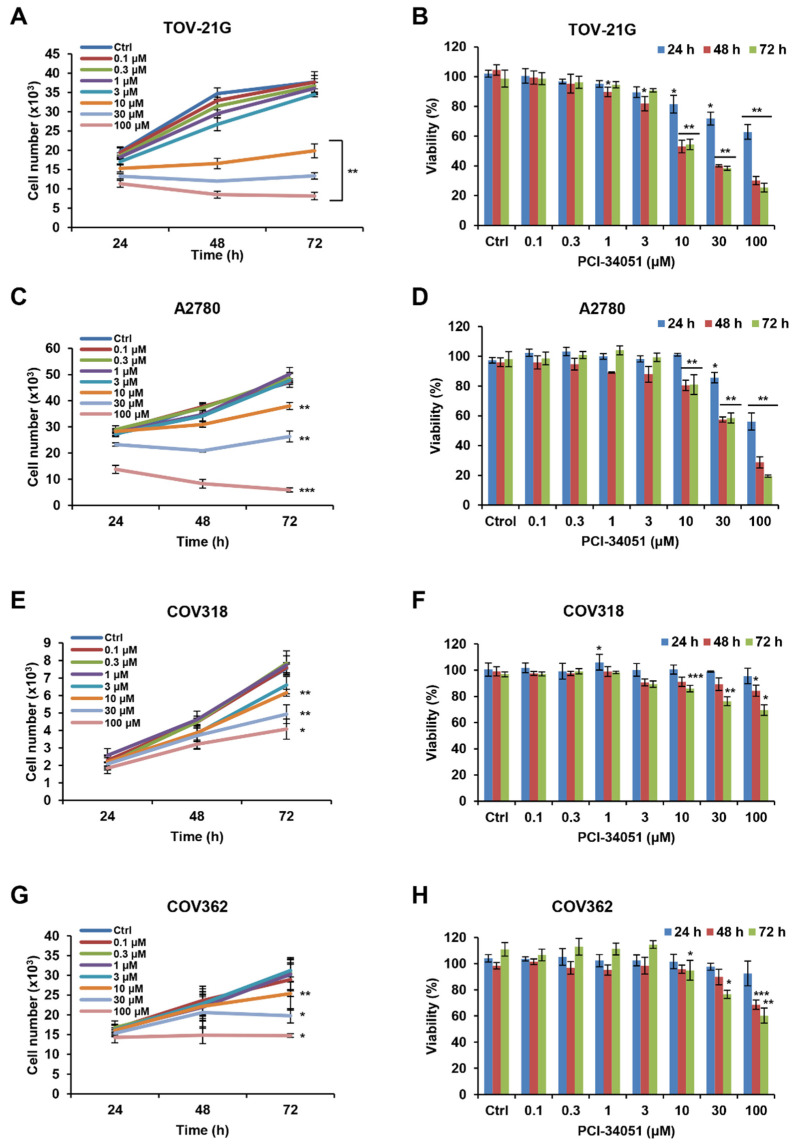
PCI-34051 suppresses the cell growth and viability of wild-type p53 ovarian cancer cells. (**A**,**C**,**E**,**G**) Cell growth and (**B**,**D**,**F**,**H**) viability of wild-type p53 ovarian cancer cell lines: (**A**,**B**) TOV-21G and (**C**,**D**) A2780 and mutant p53 ovarian cancer cell lines: (**E**,**F**) COV318 and (**G**,**H**) COV362. Cells were cultured with 0.1% DMSO (control) or PCI-34051 at the indicated concentrations for 24, 48, and 72 h. Cell growth and viability were measured using the CCK-8 assay. Data are presented as the mean ± SD (*n* = 3). * *p* < 0.05, ** *p* < 0.01, or *** *p* < 0.001 vs. DMSO control.

**Figure 2 ijms-23-08645-f002:**
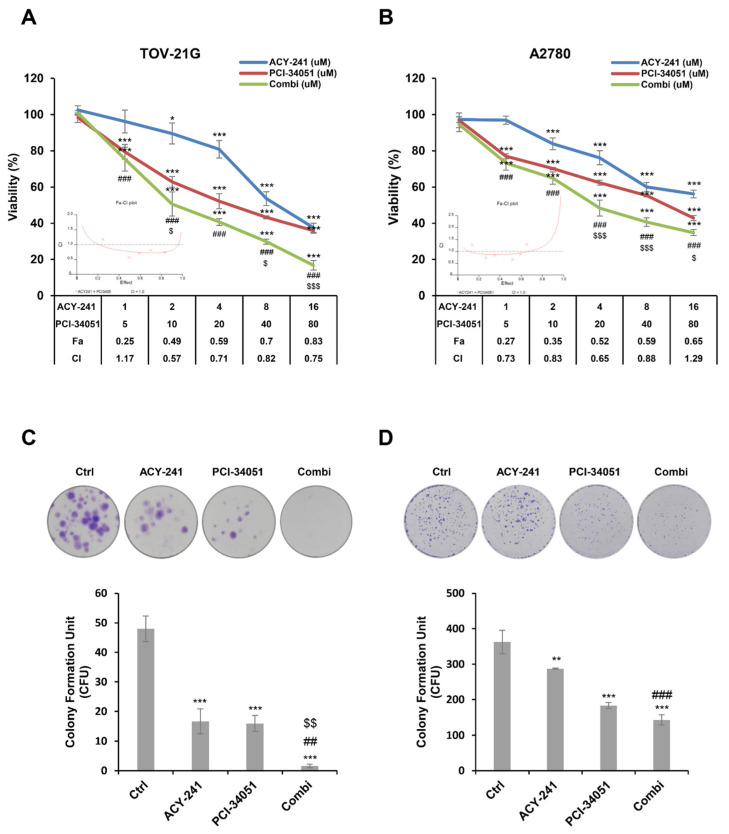
ACY-241 and PCI-34051 treatment synergistically suppresses short-term and long-term cell proliferation. (**A**,**B**) Cell viability of TOV-21G and A2780 cells treated with ACY-241 and PCI-34051 alone or in combination at a ratio of 1:5 for 48 h. Combination index (CI) values were calculated to determine the synergism of ACY-241 and PCI-34051. CI values less than 1 indicate synergism. (**C**,**D**) Colony formation in TOV-21G and A2780 cells treated with ACY-241 and PCI-34051 alone or in combination. TOV-21G cells were treated with 0.3 µM ACY-241 and 2 µM PCI-34051 and incubated for 14 days. A2780 cells were treated with 0.6 µM ACY-241 and 4 µM PCI-34051 and incubated for 7 days. Data are presented as the mean ± SD (*n* = 3). * *p* < 0.05, ** *p* < 0.01, or *** *p* < 0.001 vs. DMSO control; ^##^ *p* < 0.01 or ^###^ *p* < 0.001 vs. ACY-241-treated group; ^$^ *p* < 0.05, ^$$^ *p* < 0.01, or ^$$$^
*p* < 0.001 vs. PCI-34051-treated group.

**Figure 3 ijms-23-08645-f003:**
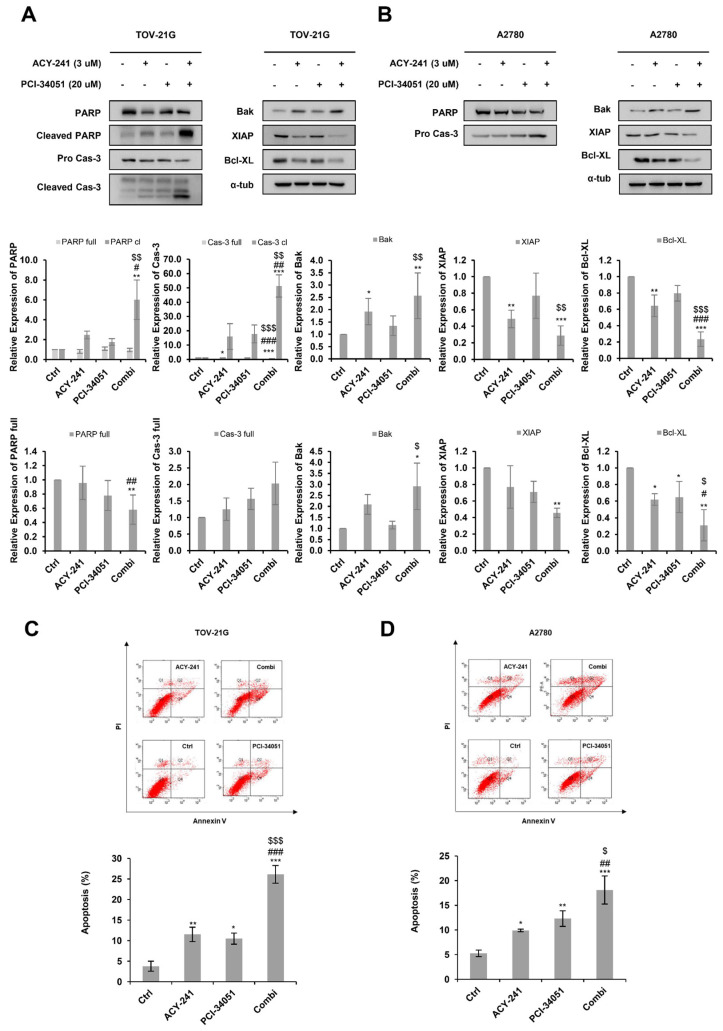
ACY-241 and PCI-34051 treatment synergistically induces apoptosis. (**A**,**B**) Immunoblotting of pro-apoptotic markers and anti-apoptotic markers in TOV-21G and A2780 cells treated with 3 µM ACY-241 and 20 µM PCI-34051 alone or in combination for 24 h. Protein expression levels were semi-quantified relative to the loading control, α-tubulin. (**C**,**D**) Apoptosis analysis of ACY-241 and PCI-34051 in TOV-21G and A2780 cells. Cells treated with 0.2% DMSO, ACY-241 (3 μM), and PCI-34051 (20 μM) alone or in combination for 48 h were double−stained with Annexin V and PI and analyzed by flow cytometry. Data are presented as the mean ± SD (*n* = 3). * *p* < 0.05, ** *p* < 0.01, or *** *p* < 0.001 vs. DMSO control; ^#^ *p* < 0.05, ^##^ *p* < 0.01, or ^###^ *p* < 0.001 vs. ACY-241-treated group; ^$^ *p* < 0.05, ^$$^ *p* < 0.01, or ^$$$^ *p* < 0.001 vs. PCI-34051-treated group.

**Figure 4 ijms-23-08645-f004:**
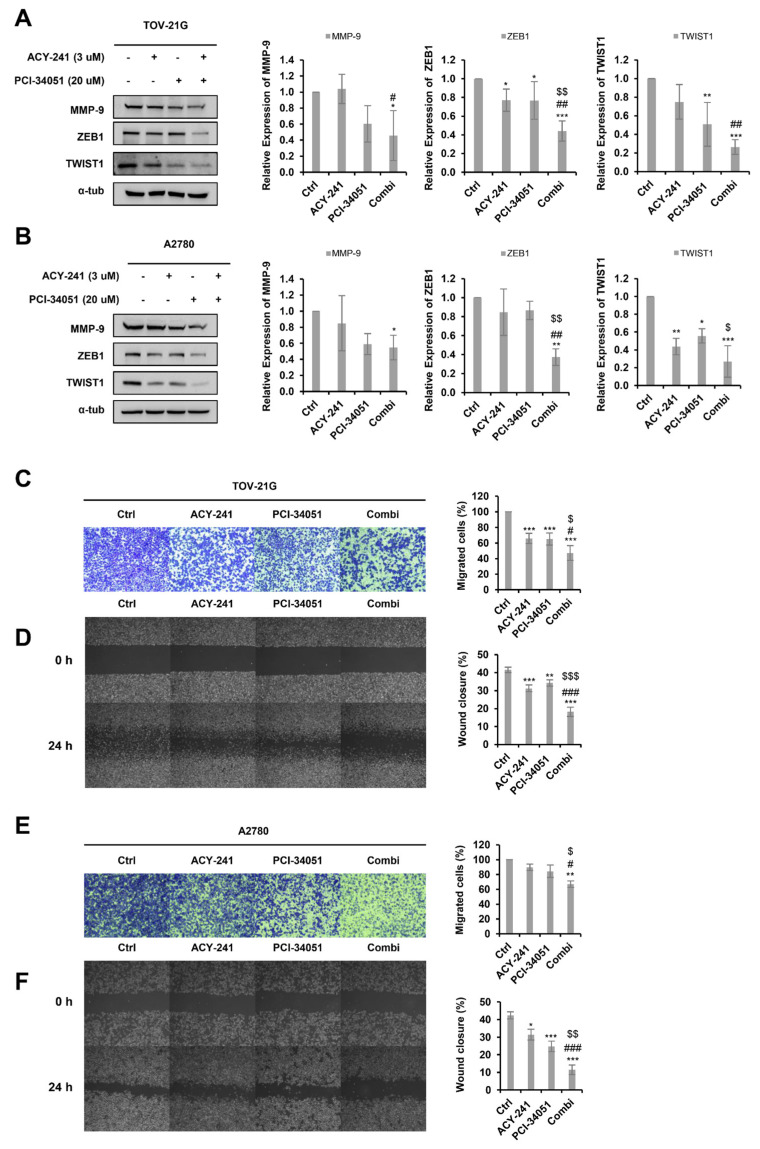
ACY-241 and PCI-34051 treatment synergistically suppresses cell migration. (**A**,**B**) Immunoblotting of TWIST1, MMP-9, and ZEB1 in TOV-21G and A2780 cells treated with 3 µM ACY-241 and 20 µM PCI-34051 alone or in combination for 24 h. Protein expression levels were semi-quantified relative to the loading control, α-tubulin. (**C**,**E**) Transwell migration assay in TOV-21G and A2780. Cells were treated with 3 µM ACY-241 and 20 µM PCI-34051 alone or in combination for 48 h. Migrated cells were stained and photographed at 100× magnification. (**D**,**F**) Wound healing assay in TOV-21G and A2780. Cells were treated with 3 µM ACY-241 and 20 µM PCI-34051 alone or in combination for 24 h. The width of the scratch was calculated and photographed at 50× magnification. Data are presented as the mean ± SD (*n* = 3). * *p* < 0.05, ** *p* < 0.01, or *** *p* < 0.001 vs. DMSO control; ^#^
*p* < 0.05. ^##^ *p* < 0.01, or ^###^ *p* < 0.001 vs. ACY-241-treated group; ^$^ *p* < 0.05, ^$$^ *p* < 0.01, or ^$$$^ *p* < 0.001 vs. PCI-34051-treated group.

**Figure 5 ijms-23-08645-f005:**
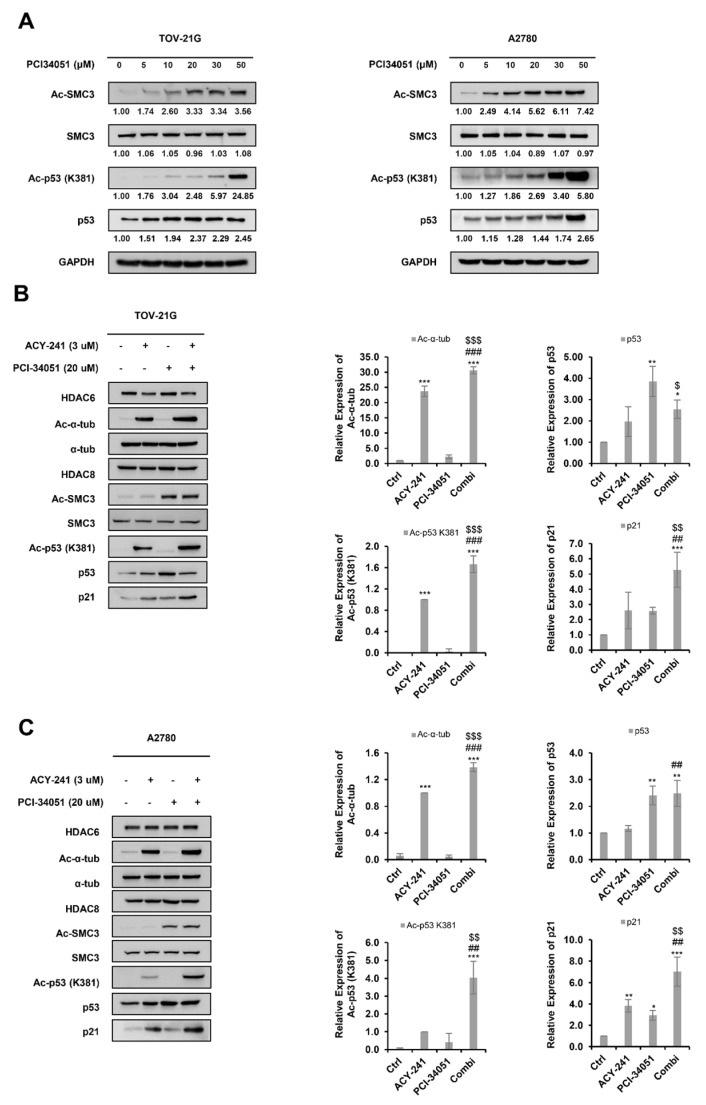
ACY-241 and PCI-34051 treatment synergistically enhances p53 stability in p53 wild-type ovarian cancer cells. (**A**) Immunoblotting of acetyl-p53. Cells were treated with indicated doses of PCI-34051 for 24 h. (**B**,**C**) Immunoblotting of acetyl-p53 and p21. Cells were treated with 3 µM ACY-241 and 20 µM PCI-34051 alone or in combination for 24 h. Protein expression levels were semi-quantified relative to the loading control, α-tubulin. Relative protein levels of acetyl-p53 were semi-quantified relative to total p53 levels and relative protein levels of acetyl SMC3 were semi-quantified relative to total SMC3 levels. Data are presented as the mean ± SD (*n* = 3). * *p* < 0.05, ** *p* < 0.01, or *** *p* < 0.001 vs. DMSO control; ^##^ *p* < 0.01, or ^###^ *p* < 0.001 vs. ACY-241-treated group; ^$^
*p* < 0.05, ^$$^ *p* < 0.01, or ^$$$^ *p* < 0.001 vs. PCI-34051-treated group.

**Table 1 ijms-23-08645-t001:** IC_50_ and GI_50_ values of PCI-34051 in ovarian cancer cell lines.

Time	72 h			
Cell line	TOV-21G	A2780	COV318	COV362
^1^ IC_50_ (µM)	9.7	28.3	127.6	120.4
^2^ GI_50_ (µM)	9.3	25.9	127.2	81.6

^1^ IC_50_: half-maximal inhibitory concentration; ^2^ GI_50_: half-maximal growth inhibition concentration.

## Data Availability

Not applicable.

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
