# Peer review of "HDAC8-Selective Inhibition by PCI-34051 Enhances the Anticancer Effects of ACY-241 in Ovarian Cancer Cells"

_ijms, 2022, doi:10.3390/ijms23158645_

Round 1

Reviewer 1 Report

Dear Editor

I have revised the work: HDAC8-selective Inhibition by PCI-34051 Enhances Anticancer Effects of ACY-241 in Ovarian Cancer Cells which is very interesting, however, I have some suggestions

1.- It is not clear why the PCI-34051 effectively suppresses cell proliferation in wild-type p53 than on mutant protein. 

2.- I suggest to include some anti-ovarian cancer drugs reported elsewhere to compare the these HDAC inhibitors tested.

3.- Due to there are pan-HDAC inhibitors, I suggest to include one of this, particularly, one that target HDAC6 and HDAC8, ej. droxinostat.

4.- It is not clear why the authors select as target HDAC6 and HDAC8 due to there are other HDAC overexpresed in ovarian cancer: https://m.ejgo.org/Synapse/Data/PDFData/1114JGO/jgo-19-185.pdf, https://molecular-cancer.biomedcentral.com/articles/10.1186/s12943-018-0855-4

Reviewer 2 Report

In this study, Kim et al. show that inhibition of HDAC8 acts synergistically with HDAC6 inhibition in p53 wild type ovarian cancer cell lines. The results are clear and relevant, although the conclusions can only be applied to cell lines.

The authors should include experiments showing the effect of ACY-241 and PCI-34051 in p53 mutant cell lines, looking at cell proliferation and viability.

The effects seem to be more marked in TOV-21G than in A2780, despite both cell lines being p53 wild type. Do the authors have any explanation for those differences?

As a general comment, the font size in the figures is small. It should be changed to increase the readability of the manuscript.

Round 2

Reviewer 1 Report

Dear Editor

I have revised the replay response and the final article’s version. I suggest to add

references  to support your responses in case of no including experimental assays. 

Reviewer 2 Report

The authors have addressed all my comments. I recommend publication of the article.

Author Response

We thank the reviewer for recognizing and reviewing the potential importance of our findings.